# Effect of Scalp Nerve Block Combined with Intercostal Nerve Block on the Quality of Recovery in Patients with Parkinson’s Disease after Deep Brain Stimulation: Protocol for a Randomized Controlled Trial

**DOI:** 10.3390/brainsci12081007

**Published:** 2022-07-29

**Authors:** Ying Sheng, Hui Wang, Xinning Chang, Peipei Jin, Shengwei Lin, Shuang Qian, Jian Xie, Wenbin Lu, Xiya Yu

**Affiliations:** 1Faculty of Anesthesiology, Changhai Hospital, Naval Medical University/Second Military Medical University, PLA, Shanghai 200433, China; shengyingno1@126.com (Y.S.); whuimz@sina.com (H.W.); xinningchang@163.com (X.C.); kingpei89@163.com (P.J.); linswdr@163.com (S.L.); 13621786541@163.com (S.Q.); xiejian1024@smmu.edu.cn (J.X.); 2Department of Anesthesiology and Perioperative Medicine, Shanghai Fourth People’s Hospital, School of Medicine, Tongji University, Shanghai 200434, China; 3Shanghai Key Laboratory of Anesthesiology and Brain Functional Modulation, Shanghai 200434, China; 4Translational Research Institute of Brain and Brain-Like Intelligence, Shanghai Fourth People’s Hospital, School of Medicine, Tongji University, Shanghai 200434, China; 5Clinical Research Center for Anesthesiology and Perioperative Medicine, Tongji University, Shanghai 200434, China

**Keywords:** deep brain stimulation, intercostal nerve block, Parkinson’s disease, postoperative analgesia, quality of recovery, scalp nerve block

## Abstract

Background: Parkinson’s disease (PD) patients who receive deep brain stimulation (DBS) have a higher risk of postoperative pain, which will affect their postoperative quality of recovery (QoR). Scalp nerve block (SNB) and intercostal nerve block (ICNB) can alleviate postoperative pain, yet their effect on postoperative QoR in PD patients has proven to be unclear. Therefore, we have aimed to explore the effect of SNB paired with ICNB on postoperative QoR. Methods: To explore the effect, we have designed a randomized controlled trial in which 88 patients with PD will be randomly assigned to either an SNB group or control group, receiving either SNB combined with ICNB or without before surgery. The primary outcome will be a 15-item QoR score at 24 h after surgery. The secondary outcomes will include: 15-item QoR scores at 72 h and 1 month after surgery; the numeric rating scale pain scores before discharge from the postanesthesia care unit (PACU) at 24 h, 72 h, and 1 month after surgery; rescue analgesics; nausea and vomiting 24 h after operation and remifentanil consumption during operation; emergence agitation; the duration of anesthesia and surgery; time to respiratory recovery, time to response, and time to extubation; the PACU length of stay; as well as adverse events. Proposed protocol and conclusion: Our findings will provide a novel method for the management of recovery and acute pain after DBS in PD patients. This research was registered at clinicaltrials.gov NCT05353764 on 19 April 2022.

## 1. Introduction

Parkinson’s disease (PD) is a prevalent neurodegenerative disease characterized mostly by motor symptoms such as tremor, rigidity, postural instability, bradykinesia, and postural reflex impairment [1,2]. Moreover, non-motor symptoms such as autonomic impairment and cognitive dysfunction may occur prior to the motor syndrome [3,4]. More than 6.1 million people experience multisystem symptoms of PD globally [5]. Although drug therapy can alleviate motor symptoms, there have been few effective treatment strategies for advanced PD [6]. Given the side effects of drug therapy, deep brain stimulation (DBS) is the most well established and reliable therapy for the motor fluctuations and symptoms in patients with advanced PD [7].

Most evidence has shown that DBS surgery can reduce PD-related pain [8,9,10]. However, PD patients undergoing DBS have a higher risk of perioperative complications, including postoperative pain which can worsen the recovery of patients, thus resulting in longer hospital stays and higher hospital costs [11,12]. A previous study has shown that analgesic methods were administrated less frequently for neurosurgery than other operations [13], and clinicians often fail to provide early sufficient postoperative analgesia due to the early assessment of consciousness. Recent studies have shown that up to 80% of patients suffered moderate-to-severe pain following neurosurgery [14,15,16]. So far, few studies have explored effective analgesic methods for DBS surgery.

Recently, opioid consumption in patients with moderate-to-severe postoperative pain has increased gradually, which has been followed by adverse events such as nausea, vomiting, respiratory depression, itchy skin, and drug dependency [17]. Therefore, multiple clinical studies have adopted nerve blocks instead of opioids to treat acute postoperative pain following craniotomy [18,19]. A previous study has shown that scalp nerve block (SNB) with ropivacaine can provide perioperative analgesia and reduce pain-related complications [20]. In addition, acute pain after thoracotomy has been found to be reduced by intercostal nerve block (ICNB) [21]. Therefore, ICNB may relieve postoperative pain in DBS surgery due to the site of the anterior chest incision. However, there is a shortage of evidence to show that SNB combined with ICNB reduces pain and improves the quality of recovery (QoR) after DBS surgery.

The SNB—the blockage of nerves including the greater occipital nerve, superficial temporal nerve, trochlear nerve, and supraorbital nerve—was developed because of its potential advantages for powerful regional anesthesia administration. Moreover, it encourages the advancement of precise neurosurgery, functional, and micro-neurosurgery [22]. The ICNB was used to provide effective postoperative analgesia in chest surgery [21,23]. The incision of DBS surgery involves the distribution of scalp nerves and intercostal nerves; thus, SNB combined with ICNB may relieve postoperative pain in patients receiving DBS surgery.

The 15-item QoR (QoR-15) questionnaire is an outcome questionnaire reported by patients that assesses the quality of recovery following surgery and anesthesia [24]. The QoR-15 score includes five areas of health status reported by patients in order to provide a meaningful assessment of their total recovery experience: pain, physical comfort, physical independence, psychological state, and emotional state. It is a useful instrument in clinical studies or trials due to the measurement errors and reliability, and a higher QoR-15 score indicates better postoperative recovery.

The present study’s main objective is to evaluate the clinical efficacy of SNB combined with ICNB for the QoR after DBS surgery. Furthermore, the present study may provide a novel method for enhanced recovery and early prevention and treatment of acute pain after DBS surgery. We hypothesize that SNB combined with ICNB can improve the postoperative QoR of patients with PD. This is based on the clinical evidence that the involved region of the scalp and intercostal nerves can be blocked by SNB combined with ICNB, providing an extensive block and better analgesia.

We have designed a single-center, prospective, randomized controlled trial including 88 PD patients scheduled for DBS surgery. The primary outcome will be a QoR-15 score at 24 h after surgery. The secondary outcomes will include: QoR-15 scores at 72 h and 1 month after operation; the numeric rating scale (NRS) pain scores before discharge from the postanesthesia care unit (PACU) at 24 h, 72 h, and 1 month following surgery; rescue analgesics; nausea and vomiting 24 h following surgery and remifentanil consumption during operation; emergence agitation; the duration of anesthesia and surgery; time to respiratory recovery, time to response, and time to extubation; PACU length of stay (LOS); as well as nerve-block-related adverse events (AEs).

## 2. Materials and Methods

The study protocol has been completed and is in accordance with the SPIRIT PRO extension reporting guidelines [25]. Figure 1 depicts the study’s patient flow diagram, and Table 1 depicts the study schedule.

### 2.1. Study Design and Setting

This is a single-center, prospective, randomized, controlled trial that is designed to investigate the efficacy of the combination of ICNB and SNB using ropivacaine for QoR after the performance of surgery on patients with PD. From April 2022 to March 2023, patients with PD who receive elective DBS surgery will be recruited in Shanghai Changhai Hospital, the Naval Medical University.

Protocol changes are not expected. The inclusion and exclusion criteria are fairly well established; however, when a trial adjustment is required, any modification in the methodology or criteria will be reported to the whole research group in a conference. The final manuscript for journal submission will include all revisions.

#### 2.1.1. Primary Outcome

The study’s primary outcome is the scores of QoR-15 at 24 h after surgery.

#### 2.1.2. Secondary Outcomes

The scores of QoR-15 at 72 h and 1 month after surgery.The NRS scores before discharge from PACU at 24 h, 72 h, and 1 month after surgery; rescue analgesics; nausea and vomiting 24 h after surgery; remifentanil consumption during operation; emergence agitation; PACU LOS; and SNB-related adverse events.The duration of anesthesia and surgery, time to respiratory recovery, time to response, and time to extubation.

#### 2.1.3. Outcome Measures

The QoR-15 questionnaire will be used to measure the patients’ quality of recovery at 24 h, 72 h, and 1 month after operation. A higher QoR-15 score (from 0 to 150) suggests greater recovery after surgery. The NRS score will be used to measure postoperative pain, and a higher score indicates more severe pain after surgery. The NRS scores will be recorded before discharge from PACU at 24 h, 72 h, and 1 month following surgery. Neuropathic pain will be assessed before surgery by a modified Douleur Neuropathique 4 (DN4) questionnaire on a scale of 0–10. A total DN4 score of more than or equal to 4 indicates neuropathic pain.

Emergence agitation is identified as improper behavior that presents itself as excitation, agitation, and concurrent orientation disorder during the anesthesia recovery phase. Respiratory recovery time refers to the interval between the end of anesthetic drugs and respiratory recovery. Response time is defined as the interval between the cessation of anesthetic drugs and response to a verbal command. Duration of time between the cessation of anesthetic drugs and extubation is referred to as the extubation time.

#### 2.1.4. Sample Size and Recruitment

The study’s primary outcome is the scores of QoR-15 at 24 h after surgery. Our preliminary study showed that the scores of QoR-15 at 24 h following DBS operation in PD patients without nerve block were equivalent to 110 (10.1). A previous study has shown that the QoR-15 score change of 8 points was deemed to be a clinically significant difference [26]. We calculated the required number of patients using PASS 11 software (NCSS, Kaysville, UT, USA) with an alpha error of 0.05 and power 95% (test for two proportions, two-sided *t*-test), and determined that 35 patients would need to be included in each group. Considering a 20% dropout rate, we defined a sample size of 88 participants.

Recruitment will be performed during a 12-month period. The study will end when the last patient enrolled undergoes an assessment of QoR 1 month after surgery. The protocol has been initiated as planned in April 2022 and will be completed in March 2023.

### 2.2. Selection of Subjects

#### 2.2.1. Participants

Participants who will undergo elective DBS surgery will be informed of the trial before surgery. Interested participants can contact the research coordinator to acquire more information about the study purpose and protocol. The clinical research coordinator will interview potential subjects to assess whether they meet the inclusion and exclusion criteria. Written informed consent will be voluntarily provided by eligible participants to the research coordinator before randomization.

#### 2.2.2. Inclusion and Exclusion Criteria

Inclusion criteria:Aged ≥ 18;American Society of Anesthesiologists (ASA) physical status of I–III;Able to communicate normally.

Exclusion criteria:Allergy to local anesthetics;Pre-existing infection at block site;Severe coagulopathy;Pre-operative neuropathic pain state;DBS surgery history;Failing to give informed consent or lack of compliance.

### 2.3. Interventional Methods

#### 2.3.1. Intervention

Participants randomized to the SNB group will receive general anesthesia combined with SNB and ICNB using 0.5% ropivacaine, which will be performed exclusively by an attending anesthesiologist. Meanwhile, patients in the control group will receive general anesthesia without the nerve blocks. An attending anesthesiologist will select the sites of the nerve blocks based on the surgical incision site. After negative aspiration, the scalp nerve will be blocked, including the greater occipital nerve (2–3 mL), superficial temporal nerve (2–3 mL), trochlear nerve (2–3 mL), and supraorbital nerve (2–3 mL). The total volume in SNB will not exceed 10 mL.

Ultrasound-guided unilateral ICNB will be conducted at the plane of T4–T5 next to the sternum in patients who are in the supine position. A longitudinally oriented, linear ultrasonic probe will be used to determine the pleura, internal intercostal muscles, rib, and innermost intercostal muscles. An echogenic needle will be inserted using an in-plane technique to target the inferior margin of rib. A volume of 10–15 mL with 0.5% ropivacaine will be injected into the intercostal spaces where the incision locates after negative aspiration. ICNB is considered successful when pleural movement is observed in intercostal spaces.

The block efficacy for formal dermatomal sensory will not be formally tested because patients under general anesthesia are unconscious. The anesthesiologist performing the nerve blocks will not visit patients in the perioperative period. After the nerve blocks, the anesthesiologist will cover the dressing at the relevant areas to hide the allocation. Only the dressing will be covered in the same site of the control group without injecting normal saline, in consideration of ethical care.

#### 2.3.2. Criteria for Termination or Adjusting the Assigned Interventions

There will be no unique criteria for changing assigned interventions, but there are situations under which the research will be terminated: (1) postoperative surgical site bleeding; (2) serious DBS surgery-related complications (such as pulmonary aspiration, wound infection, respiratory insufficiency, and delirium) and nerve-block-related AEs (such as pneumothorax and local anesthetic toxicity); (3) unexpected postoperative continuation of intubation; and (4) unanticipated reoperation 24 h following surgery.

#### 2.3.3. Anesthesia and Analgesia

Standard care will be provided for patients in two groups after arriving in the operating room, which includes pulse oximetry saturation (SpO_2_), heart rate (HR), non-invasive blood pressure (BP), and end-tidal carbon dioxide partial pressure (P_ET_CO_2_). Anesthesia will be induced with 8 mg of dexamethasone, sufentanil 0.3–0.5 μg kg^−1^, propofol 2–3 mg kg^−1^, and rocuroium 0.6 mg kg^−1^ when peripheral vascular access is obtained. After that, P_ET_CO_2_ will be maintained at 35–45 mmHg by a volume-controlled mechanical ventilation mode with a tidal volume 6–8 mL kg^−1^, and the respiratory rate 12–15 breaths per minute. Maintenance of anesthesia through continuous infusion of remifentanil (0.1–0.2 μg kg^−1^ min^−1^) and propofol (6–8 mg kg^−1^ h^−1^) will be conducted to maintain stable hemodynamics after anesthesia induction. In total, 10 µg of sufentanil will be provided 30 min before the end of surgery, and propofol and remifentanil infusion will be discontinued after surgery.

Corresponding treatment will be administered if mean arterial pressure exceeds 20% of the baseline value. The total dosage of analgesics and anesthetics will be assessed following surgery. After surgery, the anesthesiology will transfer patients to the PACU without providing patient-controlled analgesia. In total, 40 mg of parecoxib sodium will be administered as rescue treatment for patients whose NRS score is higher than 4 in the PACU or ward. If patients experience nausea or vomiting in the PACU or on discharge to the ward, they will be administered 4 mg of ondansetron.

#### 2.3.4. Adverse Event

All AEs will be recorded through the symptoms reported by the patients at each visit. Nerve-block-related AEs might include hematoma and infection in the block site, and pneumothorax and local anesthetic toxicity. If a patient is suspected to be affected by local anesthetic poisoning, an immediate stop to the nerve blocks and an active rescue treatment will be administered.

### 2.4. Randomization and Blinding

Patients will be randomly assigned to the SNB group and control group in a ratio of 1:1 using random number tables generated by a computer, including numbers 1–88 which is performed by an anesthesia nurse. The patients corresponding to the first 44 random numbers will be assigned to the control group, while the rest will be allocated to the SNB group. An anesthesia nurse with no direct clinical role in the trial will type the group assignment on separate papers, fold them up, and put them inside sequentially numbered sealed opaque envelopes.

Attending anesthesiologists who are not involved in the follow-up will perform SNB combined with ICNB after the induction of general anesthesia. Participants will be informed as to whether they will receive nerve blocks or not by the research coordinator, who will not participate in treatment or evaluation. Therefore, the attending anesthesiologists, research coordinator, and anesthesia nurse will know the allocation schedule. Group assignment will be concealed from patients, anesthesiologists responsible for operation, and investigators who are in charge of assessing and collecting data. Group allocation will not be disclosed until all data collection is completed. After consultation with the principal investigator (PI), the blinding will be removed if a concern regarding local anesthetic toxicity arises.

### 2.5. Data Collection and Analysis

#### 2.5.1. Data Collection and Management

Researchers will explain the advantages of participating in the study to participants and their authorized surrogates before DBS surgery. A specific masked researcher will collect the patients’ basic characteristics before surgery. After the surgery, the blinded investigator will conduct a follow-up to record QoR-15 scores at 24 h and 72 h after surgery in the ward, and at 1 month after surgery in the outpatient clinic. Participants will be excluded if they are unable to complete the primary outcome measure. The anesthesiologists in charge will record the patients’ intraoperative data. Other data on secondary outcomes will be collected by a blinded investigator. The data collected will be first completed on a paper Case Report Form (CRF). Following this, the data manager will merge the double data into an electric database which will be examined by two inspectors.

#### 2.5.2. Data Monitoring

The Data and Safety Monitoring Board (DSMB) of the Clinical Research Center in Changhai Hospital, which is made up of independent clinical specialists and statisticians with access to unblinded data, will monitor the study’s safety. Performance and safety of the study will be reviewed monthly by the DSMB, which is separate from the investigational site, the sponsor, and any conflicting interests. The criteria for terminating the study for a given trial participant have been described previously. The allocated intervention for participants will be revealed by the DSMB, which will also make the final decision to end the study.

The complete, anonymous final data set will be accessible to the data manager in Changhai Hospital’s Clinical Research Center. Others will have access to the final data set following the approval of the DSMB in the Clinical Research Center.

#### 2.5.3. Confidentiality

The PI will obtain all personal information and sign a confidential disclosure agreement before the trial. Each participant will be given a special research identifier to decrease the probability of accidental disclosure of identifiable data during the study. The printed materials will be secured in a locked cabinet, which is locked in the research office, and the electronic data will also be stored in a passcode-protected database in order to protect confidentiality. Access to personal data will be limited by the specific privilege assignments in the trial. Any information and documents used for assessment or statistical analyses will be represented by code.

#### 2.5.4. Statistical Analysis

Masked statistical analysts will conduct the analyses at the Clinical Research Center of Changhai Hospital. The full analysis set including all randomized patients and per protocol set, including patients with the primary outcome, will be performed. Safety and efficacy analyses will be conducted based on the intention-to-treat analysis population, which includes all randomized patients. Missing data will be handled using the mixed model for repeated measurements.

All statistical analysis will be performed by SPSS 23.0 statistical analysis software (IBM SPSS, Armonk, NY, USA). Continuous variables in normally distributed data represented as mean (SD) will be compared using the Student’s t-test, while continuous variables in non-normal distribution data reported as medians (quartiles) will be compared using Mann-Whitney *u* test. Categorical variables described as frequency and percentages will be compared using χ^2^ test or Fisher’s exact test. *p* < 0.05 will be regarded as statistically significant.

## 3. Discussion

The study will investigate whether SNB combined with ICNB can improve the postoperative quality of recovery in PD patients. Although nerve blocks are a simple strategy that can relieve pain after a craniotomy, its effect on the quality of recovery after surgery has not been studied scientifically in a randomized controlled study. In this study, we assume that the patients with SNB and ICNB have a higher score of QoR-15 than the control group 24 h after surgery. The secondary aim is to determine whether SNB and ICNB can have a beneficial effect on postoperative pain and postoperative nausea and vomiting (PONV), as well as other anesthesia-related events. Patients are randomly divided into an SNB group and control group, to which clinical investigators and assessors will be blinded at each follow-up. To the best of our knowledge, this is the first study to explore the impact of SNB combined with ICNB on the quality of recovery and will provide a new insight into the management of acute pain in PD patients receiving DBS surgery.

Our preliminary study has found that more than 70% of patients experienced moderate-to-severe pain after DBS, similar to previous studies [14,15,16]. However, long-term postoperative pain affects the quality of recovery in PD patients, which will increase complications and prolong hospital stays. Given its reputed analgesic benefits, SNB combined with ICNB offers investigators an exciting opportunity to potentially make a significant impact on patient-centered postoperative outcomes for PD patients with DBS surgery.

In this protocol, the attending anesthesiologist is not masked to the group assignment because they will be required to conduct the nerve block. However, researchers masked to the group allocation will evaluate the primary outcome. In addition, formal dermatomal evaluation of block efficacy will not be conducted because nerve blocks will be performed under general anesthesia. This increases the risk that certain blocks may not be completely effective. Scalp nerves are superficial and easy to block, and ICNB performed under ultrasound guidance is consistent with routine clinical practice. Thus, our results should still be generalizable to widespread clinical practice. Lastly, this study is a single-center study, which may result in bias. Nonetheless, the findings of the present study may promote the development of enhanced postoperative recovery and popularize the optimal pain management of patients with DBS surgery.

## Figures and Tables

**Figure 1 brainsci-12-01007-f001:**
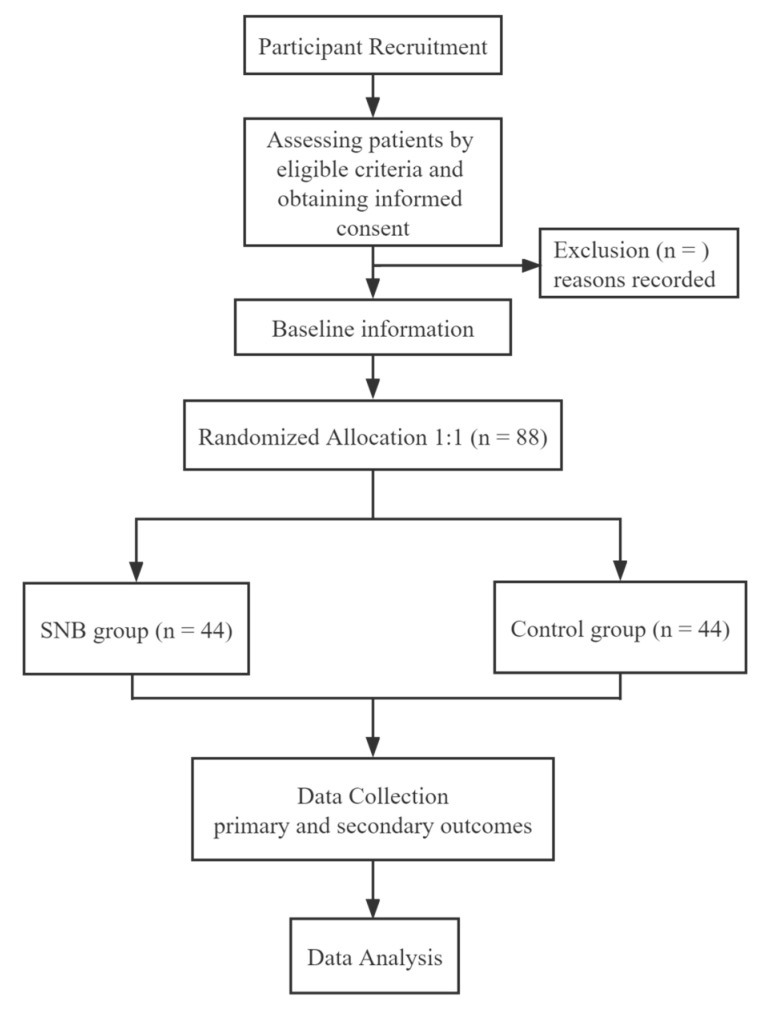
Flow chart of participants in the trial.

**Table 1 brainsci-12-01007-t001:** The schedule of enrollment, interventions, and assessments for the trial.

		Enrollment	Allocation	Postallocation					Postoperative		
Time point				Induction	Postinduction	Operation	PACU	PACU discharge	24 h	72 h	1 m
Enrollment		**×**									
Eligibility screen		**×**									
Informed consent		**×**									
Allocation			**×**								
Basic information		**×**									
Interventions											
SNB group	Blocking				**×**						
	Dressing				**×**						
Control group	Blocking										
	Dressing				**×**						
Assessment											
QoR-15 scores									**×**	**×**	**×**
NRS scores								**×**	**×**	**×**	**×**
PONV											
Remifentanil consumption											
Rescue analgesics											
Emergence agitation							**×**				
Anesthesia duration											
Operation duration						**×**					
Respiratory recovery time							**×**				
Response time							**×**				
Extubation time							**×**				
PACU LOS							**×**				
DN4		**×**									
AEs					**×**	**×**	**×**	**×**	**×**	**×**	**×**

SNB, scalp nerve block; QoR, quality of recovery; NRS, numeric rating scale; PONV, postoperative nausea and vomiting; PACU, postanesthesia care unit; LOS, length of stay; DN4, Douleur Neuropathique 4; AEs, adverse events.

## Data Availability

The data supporting the findings of this study will be obtained from the corresponding author according to reasonable request, and the corresponding author/s can be directly contacted for further inquiry.

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
