# Peer review of "Effect of Scalp Nerve Block Combined with Intercostal Nerve Block on the Quality of Recovery in Patients with Parkinson’s Disease after Deep Brain Stimulation: Protocol for a Randomized Controlled Trial"

_brainsci, 2022, doi:10.3390/brainsci12081007_

Round 1
Reviewer 1 Report
Review of a manuscript “Effect of Scalp Nerve Block Combined with Intercostal Nerve Block on Quality of Recovery in Patients with Parkinson's Disease after Deep Brain Stimulation: Protocol for a Randomized Controlled Trial “ by Ying Sheng et al., submitted to “Brain Science”.
Parkinson’s disease is prevalent severe neurodegenerative diseases associated with motor symptoms and non-motor for which there is no efficient treatment affecting the main course of the disease. Deep brain stimulation is one of the approaches that may alleviated the motor fluctuations and other symptoms of this disorder. The authors evaluated the efficiency of scalp nerve block (SNB) and intercostal nerve block (ICNB) in post-operative period after deep brain stimulation. They also assessed its efficacy of these methods in early prevention and treatment of acute pain after deep brain stimulation surgery. This is an important biomedical field and the results presented in the manuscript will be interesting for the readership of “Brain Science”.
The following corrections and additions should be made.
Abstract:
In the Abstract the authors wrote “Results and conclusion”. However, The Results section is absent in the manuscript. This is because the manuscript proposes a controlled trial of patients with PD scheduled for deep brain stimulation surgery. So the authors should not use the word Results, and replace it on another term (proposed protocol or something similar).
Introduction
Line 45: “Although drug therapy can alleviate motor symptoms, there were few effective treatment strategies for advanced PD”. The authors should add here a recent review on PD: ”Biomarkers in Parkinson’s Disease”. Chapter in a book. Peplow P.V., Martinez B., Gennarelli T.A. (eds) Neurodegenerative Diseases Biomarkers. 2022. Neuromethods, vol 173. pp 155-180. Humana, New York, NY. https://link.springer.com/protocol/10.1007/978-1-0716-1712-0_7
Line 46: ”Given the side effects of drug therapy and disease progression…” This is an awkward sentence, “disease progression” is not in a right place here. Deletion will be beneficial.
Line 50: ”A great many of evidence have shown…” should be replaced on “Many evidence…”
Materials and Methods
Figure 1: Flow chart of participants in the trial
The fonts of letters should be increased for easier reading.
Table 1
It is unclear why the authors propose to add “Interventions”, “SNB group”, “Control group”, “Block” if they did not put a mark in any of these terms.
Line 200: ”The anesthesiologist performing the nerve blocks will not be involved in the present study …” The sense is unclear. How is it possible that the anesthesiologist will perform the nerve blocks but will not be involved in the study”?
3. Discussion
Line 301-2: “The study investigates whether the SNB combined with ICNB can improve postoperative quality of recovery in PD patients.” Should be replaced on “The study will investigate whether the SNB combined with ICNB can improve postoperative quality of recovery in PD patients.”
,
Reviewer 2 Report
The paper discusses
Effect of Scalp Nerve Block Combined with Intercostal Nerve 2 Block on Quality of Recovery in Patients with Parkinson's 3 Disease after Deep Brain Stimulation: Protocol for a Random-4 ized Controlled Trial
The authors cite Prof Bergman that developed the DBS method
Results and conclusion: the finding will provide a novel method for management of recovery and 33 acute pain after DBS in the PD patients. This study was registered at clinicaltrials.gov 34 NCT05353764 on April 19, 2022.
it is well written and contains 1 fig and 1 table as well as references it can be accepted for publication as is
Author Response
Thanks for your reviews. Certainly, this is the first trial to examine the impact of the SNB combined with ICNB on the quality of recovery, and we will provide a new insight in management of acute pain after DBS in the PD patients.